# Structure and Function of Ion Channels Regulating Sperm Motility—An Overview

**DOI:** 10.3390/ijms22063259

**Published:** 2021-03-23

**Authors:** Karolina Nowicka-Bauer, Monika Szymczak-Cendlak

**Affiliations:** 1Department of Chemical Physics, Faculty of Chemistry, Adam Mickiewicz University in Poznań, 61-614 Poznan, Poland; 2Department of Animal Physiology and Development, Faculty of Biology, Adam Mickiewicz University in Poznań, 61-614 Poznan, Poland; monikasz@amu.edu.pl

**Keywords:** sperm motility, ion channels, membrane channels, calcium, potassium, chloride, sodium, proton

## Abstract

Sperm motility is linked to the activation of signaling pathways that trigger movement. These pathways are mainly dependent on Ca^2+^, which acts as a secondary messenger. The maintenance of adequate Ca^2+^ concentrations is possible thanks to proper concentrations of other ions, such as K^+^ and Na^+^, among others, that modulate plasma membrane potential and the intracellular pH. Like in every cell, ion homeostasis in spermatozoa is ensured by a vast spectrum of ion channels supported by the work of ion pumps and transporters. To achieve success in fertilization, sperm ion channels have to be sensitive to various external and internal factors. This sensitivity is provided by specific channel structures. In addition, novel sperm-specific channels or isoforms have been found with compositions that increase the chance of fertilization. Notably, the most significant sperm ion channel is the cation channel of sperm (CatSper), which is a sperm-specific Ca^2+^ channel required for the hyperactivation of sperm motility. The role of other ion channels in the spermatozoa, such as voltage-gated Ca^2+^ channels (VGCCs), Ca^2+^-activated Cl-channels (CaCCs), SLO K^+^ channels or voltage-gated H^+^ channels (VGHCs), is to ensure the activation and modulation of CatSper. As the activation of sperm motility differs among metazoa, different ion channels may participate; however, knowledge regarding these channels is still scarce. In the present review, the roles and structures of the most important known ion channels are described in regard to regulation of sperm motility in animals.

## 1. Introduction

Sperm motility is a unique characteristic of sperm physiology that is necessary to achieve egg fertilization. Motility acquisition, maintenance, and modulation are essential for male fertility. The first steps of motility initiation vary among metazoa and are strictly correlated to the environment in which fertilization takes place. During external fertilization in aquatic animals and amphibians, the initiation depends on the osmolality of water and/or the presence of chemoattractants, termed as sperm-activating peptides (SAPs), released from egg jelly [1,2,3,4,5]. Generally, in marine fishes, the motility of spermatozoa is induced by hyperosmotic seawater, whereas in freshwater fish spermatozoa the motility is induced by hypo-osmotic fresh water (precisely revised by Reference [5]). In the case of in-water fertilization, the sperm motility differs between species and depends on the distance that spermatozoa have to travel to reach an oocyte [6]. In sessile organisms (e.g., urchins), spermatozoa have to swim a long way to reach an oocyte and their motility is of the same high activity for a long time; however, spermatozoa from swimming species (e.g., fish) are released near oocytes and the intensity of sperm motility decreases immediately after activation [6].

In mammals, where the fertilization is internal, the spermatozoa are activated during their transit through the epididymis [7], and after ejaculation, most of the sperm population is motile. The induction of motility in the epididymis is linked to the gradual transition from phosphatase to kinase activities, and therefore with the gradual phosphorylation of proteins [8]. This basic type of motility is characterized by low-amplitude symmetrical tail movements that enable linear motility for spermatozoa; however, to ensure the success of fertilization, spermatozoa must become hyperactivated after entering the uterus (reviewed in Reference [8]). During hyperactivation, sperm flagellar beating becomes more intensive, in order to enable spermatozoa to penetrate dense mucus, detach from the oviductal epithelium, and reach an oocyte [8]. Hyperactivation is a part of sperm capacitation, which comprises a series of physiological changes that enable spermatozoa to fertilize an oocyte [9].

Despite the varying locations and methods of fertilization, the activation and maintenance of sperm motility in all animals is dependent on the membrane potential (Vm), intracellular pH (pH_i_), and proper balance of intracellular ions. All of these factors are tightly interlinked, and their effective cooperation provides cellular signals required for sufficient sperm motility. Among all ions involved in sperm motility, the most important is Ca^2+^. Low intracellular Ca^2+^ ([Ca^2+^]_i_) concentrations (10–40 nM) trigger symmetrical beating of the flagellum, whereas higher concentrations (100–300 nM) induce hyperactivation of spermatozoa. Nevertheless, very high concentrations of [Ca^2+^]_i_ (about 9 µM) suppress motility (revised in Reference [10]). In spermatozoa, [Ca^2+^]_i_ plays the role of a secondary messenger that is involved in the activation of signaling pathways that regulate sperm motility. One of them is the activation of soluble adenyl cyclases (sAC), which generate most of the cyclic adenosine monophosphate (cAMP) in spermatozoa. Cyclic AMP activates serine/threonine protein kinase A (PKA), which triggers a cascade of protein phosphorylation events that lead to the induction of sperm motility [10].

The high [Ca^2+^]_i_ concentrations that are necessary for hyperactivation are ensured by alkaline intracellular environments [11]. Alkaline pH_i_ activates specialized Ca^2+^ channels (CatSper) and increases sAC activity. Notably, sAC/cAMP/PKA pathway activation stimulates protein tyrosine kinases (PTKs), which phosphorylate tyrosine on flagellar proteins related to sperm motility [12,13]. Additionally, alkaline pH_i_ levels are required by dyneins, which form peripheral doublets for axonemes and exhibit ATPase activity [14]. For extensive reviews on the factors and signaling pathways regulating sperm motility, please see the relevant literature [8,9,10,15]. Besides the regulation of sAC activity, pH_i_ and Vm regulate the ion channels that ensure ion homeostasis. In spermatozoa, this homeostasis must be modulated, to provide a force for sperm motility and successful fertilization. The maintenance of proper [Ca^2+^]_i_ levels is possible due to the clock-like cooperation of ion channels, transporters, and pumps along the sperm membrane. The activities and sensitivities of these structures directly arise from their topology, which enables them to react effectively to signals from the external and internal environments that sperm cells encounter. In this review, the structure and physiological role of some of the important ion channels that are critical to spermatozoa motility are discussed.

## 2. Calcium Channels

As mentioned above, the Ca^2+^ channels play an important role in the regulation of sperm motility as Ca^2+^ is a common secondary messenger engaged in several cellular signaling pathways. Ca^2+^ is required for sAC/cAMP/PKA pathway activation [10], as well as the maintenance of sperm mitochondrial functioning and ATP production [16], which are critical for sperm cells motility. An increase in sperm [Ca^2+^]_i_ during motility activation (primary or hyperactivation) has been documented to be present in many eukaryotic organisms, ranging from sea urchins [17], marine and freshwater fish (reviewed by Reference [5]), amphibians [1], birds [18], and mammals, including humans [19,20,21]. Although Ca^2+^ is required for sperm motility, its concentrations have to be kept at adequate levels to maintain motility [22]. The maintenance of appropriate [Ca^2+^]_i_ concentrations in sperm cells is regulated by several calcium channels.

It is noteworthy that besides Ca^2+^ channels, [Ca^2+^]_i_ concentrations in spermatozoa are also regulated by the plasma membrane Ca^2+^-ATPase (PMCA) [23], Na^+^/Ca^2+^ exchanger (NCX) [24], and K^+^-dependent Na^+^/Ca^2+^ exchanger (NCKX) [25].

### 2.1. Voltage-Gated Ca^2+^ Channel (VGCC)

Voltage-gated Ca^2+^ channels (VGCCs) are a group of voltage-gated ion channels that activate on membrane depolarization and mediate Ca^2+^ influx in response to action potential and subthreshold depolarizing signals [26]. Due to different physiological, pharmacological, and regulatory properties, there are several types of VGCCs, including L-, N-, R-, P/Q-, and T-type channels (Figure 1c). A VGCC is a heteromultimer consisting of several different subunits, including the α1 (encoded by *CACNA1* genes), α2δ (*CACNA2D*), β (*CACNB*), and γ (*CACNG*) subunits, of which α1 is a core-forming subunit (Figure 1a,b) [26]. Nevertheless, the presence of the γ subunit in certain channels is putative. Regarding the structural topologies, all VGCCs are similar but not identical in terms of their resulting properties [26]. A scheme of the VGCC structure is presented in Figure 1.

VGCCs have been found in spermatozoa from a vast range of species: newts [2], marine fish [28], and mammals, including humans [29,30,31]. According to the literature, the following types of VGCCs have been identified in sperm cells: L-type (“long-lasting”, a channel activated by high voltage that is resistant to ω-conotoxin and ω-agatoxin) [28,29,30,31], T-type (“transient”, a channel operated by low voltage) [30,32], and P/Q-type (“Purkinje”, a channel activated by high voltage that is resistant to ω-conotoxin and blocked by ω-agatoxin) [30]. In equine spermatozoa, the localization of the L-type channel has been found to be restricted to the sperm neck and principal piece of the flagellum, which confirms its engagement in sperm motility. In Atlantic salmon, VGCCs have been found to positively regulate sperm activation (T-type) and total and progressive motility (L-type) [28,32] whereas in urodeles, the channels were found to be involved in the initiation of sperm motility while in contact with eggs [2]. The activities of L-type VGCCs were also documented in bull [33] and in human spermatozoa [31]. Both the L- and T-type channels were reported to participate in human sperm acrosome reactions [31]. Interestingly, Garza-López et al. [34] documented that the presence of four aspartic acids in the pore region of an L-type VGCC ensures sensitivity and permeability of the channel to Cd^2+^ in human sperm cells. Interestingly, Córdoba and Beconi [35] have documented that VGCCs can be activated by progesterone in non-capacitated cryopreserved bovine spermatozoa.

### 2.2. Transient Receptor Potential Vanilloid (TRPV)

Transient receptor potential vanilloids (TRPVs) represent a subfamily of transient receptor potential cation channels (TRP channels) that consists of six subtypes divided into two groups based on their sensitivity to temperature and Ca^2+^- permeability, namely TRPV1/TRPV2/TRPV3/TRPV4 and TRPV5/TRPV6 [36]. Among all TRPVs, only the TRPV1–4 channels are thermosensitive (thermo-TRPV channels). They have 40–50% sequence identity and are modestly permeable to Ca^2+^ (they pass 3–12 Ca^2+^ with each Na^+^ into a cell) [36,37,38], whereas TRPV5 and TRPV6, which are highly divergent from thermo-TRPVs (only ∼30% sequence identity), are particularly homologous to each other (75% sequence identity) and highly selective for Ca^2+^ (100–130 Ca^2+^ with each Na^+^) [36,37,39]. In their various structures, TRPVs are homo- or heterotetramers. Each monomer consists of six TMs with a pore loop between five and six TM. This pore is responsible for the channel cation selectivity and the modulation of temperature activation [39]. Both the N- and C-terminal ends are oriented intracellularly and the N-terminus contains three to six ankyrin repeats. The channels can interact with phosphatidylinositol 4,5-bisphosphate (PIP2) and CaM, which modulates the channel activity. In addition, TRPVs can be activated by thermal, mechanical, and osmotic stress [37,39].

All six of these TRPV channel subtypes seem to be present in the spermatozoa of vertebrates, although TRPV1 and TRPV4 are more frequently reported. In 2013, Majhi et al. [40] reported the presence of thermo-TRPVs in sperm cells from aquatic animals for the first time. The group demonstrated the presence of TRPV1 in spermatozoa from a freshwater teleost fish (*Labeo rohita*, the carp family). TRPV1 is a heat/pH/lipid/voltage-modulated channel which becomes desensitized by internal Ca^2+^; however, it is not activated by Ca^2+^-store depletion [36]. The intensity of its current is increased by an acidic pH and is modulated by intracellular PIP2 (reviewed by Reference [36]). There are many controversies concerning the role of PIP2; however, its binding probably sensitizes the channel and increases the temperature threshold for activation [41]. It seems that the main role of TRPV1 sensing pertains to the C-terminus. It contains a TRP domain and sites for CaM, phosphoinositide, and protein kinase binding, which are involved in the regulation of voltage-gated channel opening and regulation of the temperature sensor of the receptor [37]. The N-terminus contains sites for ATP binding. The binding of Ca^2+^ and CaM displaces ATP and triggers a conformational change that leads to a closed and desensitized channel [37]. The activation of TRPV1 is positively correlated with sperm motility. In a study of de Toni et al. [42], human spermatozoa migrating towards a temperature gradient of 31–37 °C showed higher levels of TRPV1 at the protein and mRNA levels. In zebrafish, a targeted blockage of TRPV1 resulted in a suppression of sperm motility and its recovery after channel reactivation [43]. Accordingly, in *Labeo rohita*, the activation of TRPV1 by endogenous activator N-arachidonoyl dopamine (NADA) improved the quality and duration of sperm motility [40]. Experiments with the blocking and activation of TRPV1 have indicated that this ion channel not only activates basic and hyperactivated motility, but also capacitation and acrosome reactions in boar spermatozoa [44]. The immunostaining of TRPV1 has indicated this channel to be localized in the sperm head, the acrosome, neck, and flagellum [40,42,43,44]. The activation of the TRPV4 channel depends on the extracellular osmolality, mechanical stimuli (e.g., membrane stretching and shear stress), pH, and lipids [36,45]. Like TRPV1, TRPV4 is sensitized by its phosphorylation at specific sites and is modulated by CaM [37,46]. In mammalian spermatozoa, TRPV4 seems to be localized in the sperm flagellum and head [47,48]. Kumar et al. [47] demonstrated the presence of head-to-tail Ca^2+^ wave propagation in human spermatozoa upon the pharmacological modulation of TRPV4. The literature data clearly indicate the role of TRPV4 in sperm motility. Hamano et al. [48] suggested the role of the TRPV4 thermosensitive properties in mouse sperm motility, where the ratio of sperm from *Trpv4* knockout mice was lower at the high temperature level gradient than in wild-type animals. Next, the swim-up fraction of human spermatozoa displayed a higher expression level of TPRV4 than the rest of the cells [47]. Additionally, according to Mundt et al. [49], TRPV4 may be involved in hyperactivated sperm motility. The proposed mechanism suggests that the activation of TRPV4 triggers initial membrane depolarization, thereby facilitating the gating of other channels (CatSper and H_v_1) required for the induction of hyperactivation. To the best of our knowledge, the rest of the TRPVs (TRPV2, TRPV3, TRPV5, and TRPV6) have only been reported to be present in duck spermatozoa [50]. According to the authors [50], although the localizations of these TRPVs were enriched in the sperm tail, some of them had distinct localization patterns within cells, suggesting their different roles in fertilization.

### 2.3. Store-Operated Ca^2+^ Channel (SOCC)

Store-operated Ca^2+^ channels (SOCCs) in animal somatic cells are plasma membrane channels that are closely topologically and physically related with the endoplasmic reticulum (ER). SOCCs are inward rectifiers (they pass positive charges more easily in the inward direction than in the outward direction), and generally their role is to supply the ER with Ca^2+^ from an extracellular space after Ca^2+^ has been released and pumped out across the plasma membrane. In addition, SOCCs also elevate the concentration of [Ca^2+^]_i_ (reviewed in Reference [51]). As SOCCs are not voltage-dependent channels, they can be active at negative membrane potentials at which depolarization-sensitive channels (e.g., most VGCCs) are inactive. SOCCs are created by the ORAI1-3 proteins, which are encoded by ORAI1–ORAI3 genes in humans, respectively. A single channel is created by one ORAI protein. The binding of Ca^2+^ inactivates the channel [51] (see Figure 2).

Prakriya and Lewis [51] proposed a detailed mechanism of ORAI action. In case of Ca^2+^ depletion in the ER, the specific protein STIM1, which is dispersed within the ER membrane, undergoes a conformational change followed by the binding of STIM1 to the PIP2 present in the plasma membrane and trapping STIM1 at the ER and plasma membrane junction. The binding of STIM1 traps the ORAI protein and activates the channel, leading to the release of Ca^2+^ from the ER. The channel inactivates upon Ca^2+^ binding to the central loop, thereby disassociating the ORAI and STIM1 complex. Additionally, the binding of CaM might also inactivate the channel by displacing STIM1 from ORAI [51,52]. It can be suggested that in spermatozoa a SOCC’s mode of action must be modified as mature sperm cells lack an ER; however, the well-known location of Ca^2+^ storage in spermatozoa is in mitochondria [53]. A scheme of the SOCC–ORAI1 structure is presented in Figure 2.

A transcriptomic and proteomic study by Darszon et al. [54] documented the presence of the ORAI1-3 channels in mouse spermatozoa. The proteins were shown to co-localize with STIM1 (and STIM2) in the mouse sperm head and flagellum [54], whereas, in chickens, ORAI1 was found in all compartments of the sperm cells [18]. The literature data indicate that SOCCs play an important role in the regulation of sperm physiology in terms of the inhibition of SOCCs reducing sperm motility and acrosome reactions [18,24]. A similar observation was documented by Yoshida et al. [55,56] in ascidian spermatozoa, where the presence of a SOCC inhibitor reduced asymmetrical flagellar beating and turning movements, which are typical signs of sperm chemotaxis. The regulation of sperm motility and acrosome reactions can be connected with the induction of 5′ AMP-activated protein kinase (AMPK) phosphorylation [18].

### 2.4. Cation Channel of Sperm (CatSper)

The cation channel of sperm (CatSper) is the most studied sperm Ca^2+^ channel due to its sperm-specificity and essential roles in the hyperactivation of the sperm flagellum, egg chemotaxis, capacitation, and acrosome reactions (reviewed in Reference [57]). CatSper is not only permeable to Ca^2+^, but it also enables monovalent cations (Na^+^ and Cs^+^) and a bivalent cation (Ba^2+^) to enter sperm cells in case of the absence of extracellular Ca^2+^. The channel is pH-sensitive, as alkaline pH_i_ activates CatSper [57]. Additionally, its activity is also regulated by the membrane voltage, cyclic nucleotides (e.g., cAMP and cGMP), phosphorylation, active biomolecules (speract, progesterone, and prostaglandins), ZP-glycoproteins, and bovine serum albumin (BSA) [57,58].

CatSper is a heterotetrameric complex consisting of four pore-forming α subunits (CatSper1-4) and six additional auxiliary subunits: CatSperβ (beta), CatSperγ (gamma), CatSperδ (delta), CatSperε (epsilon), CatSperζ (zeta), and the EF-hand calcium-binding domain-containing protein 9 (EFCAB9), which are encoded by at least 10 genes [59,60,61,62,63]. Sequence identity between CatSper1–4 subunits is rather low and ranges from 16 to 22% [64]. Apart from differences in amino acids composition, the all α subunits differ with lengths of their loops and N- and C-termini.

According to the literature, homology of CatSper among species is also low [64]. Following comparative genomics done by Cai and Clapham [65], genes of CatSperα and CatSperβ subunits have been lost through metazoan evolution, especially in vertebrate lineages, such as Agnatha, Teleostei, Amphibia, and Aves; however, that is in complete opposition to a study by Lissabet et al. [66], where CatSper was found to be present in *Salmo salar* (the infraclass: Teleostei) sperm cells. Nevertheless, the CatSper channel has also been found in sea urchin [67] and a variety of mammalian species (human, mouse, horses, and boars) [58,68,69,70,71]. Interestingly, in equine spermatozoa, there were identified species-specific differences in structure in the histidine-rich pH-sensor region (see Figure 3a) of the CatSper1 subunit [71].

The CatSper sensitivity is linked to the environment of the female reproductive tract. Once spermatozoa are transferred to female reproductive tract, the activation of CatSper is required for the change of flagellar shape and beating frequency. This hyperactivation of sperm motility is necessary to overcome the fluids and distance to the egg. The female reproductive tract supports this activation with its alkaline environment, progesterone, prostaglandins, and ZP-glycoproteins.

According to the literature, CatSper is localized in sperm principal piece [70], and in mice, it was documented to be distributed in linear quadrilateral nanodomains along the flagellum [69]. The precise structural distribution of CatSper complexes in the flagellum has been revised by Rahban and Nef [72]. Despite flagellar localization, CatSper mediates calcium influx that propagates within seconds through the midpiece and the head after the channel activation (see review by Reference [72]).

The proper functioning of CatSper seems to be dependent on the all subunits creating the channel. The lack of any of the four CatSperα subunits disables hyperactivation of sperm motility in mice [71]. Targeted disruption of murine CatSper3 or CatSper4 prevented spermatozoa from hyperactivation but not from basic motility [70], indicating the role of CatSper in final steps of sperm journey. In humans, patients with asthenoteratozoospermia (reduced sperm motility and morphology) were diagnosed with deletions in *CatSper2* [73,74] or with an insertion in *CatSper1* genes [75]. A patient with a deficiency in CatSper function had a homozygous in-frame 6-bp deletion in exon 18 of *CatSperε* [69]. A targeted disruption of *CatSperζ* in mice interrupted the structural continuity of CatSper location along the flagellum and made the proximal piece of sperm tail inflexible making males sub-fertile [60]. Similar results were observed by Hwang et al. [62], in *Efcab9* knockout mice. Additionally, the authors demonstrated that the EFCAB9 subunit is essential for pH-dependent and Ca^2+^-sensitive activation of the CatSper channel. A scheme of the CatSper structure is presented in Figure 3.

In sea urchins, CatSper was denoted as a fundamental Ca^2+^ channel activated in spermatozoa upon speract binding, whereas other Ca^2+^ channels only had a complementary role [67]. In humans, CatSper was documented to be activated by progesterone and prostaglandins in the presence of alkaline pH_i_ [76]. According to work of Lishko et al. [76], progesterone activates Ca^2+^ current in CatSper by binding to a non-genome progesterone receptor in sperm membrane. This receptor is alpha/beta hydrolase domain-containing protein 2 (ABHD2) [77]; however, the mediation of Ca^2+^-entry by prostaglandins seems to be activated through different, not-yet-known binding sites. Additionally, not all prostaglandins have a potential to induce CatSper activity (for details, see Reference [76]). The stimulation with progesterone seems to play a crucial role not only in CatSper regulation but also in further events, following Ca^2+^ entry: activation of PTK and phosphorylation of tyrosine residues in the principal piece [78]. A single cell analysis of human spermatozoa showed reduced progesterone-sensitivity in sperm cells from sub-fertile patients that was positively correlated with fertilization rate [79]. The role of progesterone in the induction of hyperactivation was also reported in pigs [80]. The addition of progesterone induced a release of porcine spermatozoa from oviduct cells (simulation of sperm reservoir in female reproductive tract) activating CatSper channels [80]. Nevertheless, progesterone/prostaglandin-activation of CatSper is not a universal mechanisms among mammalian species and it was proved to be not present in mouse sperm [76]. Next, even in humans also other components of female genital tract, e.g., from follicular fluid, seem to induce Ca^2+^ currents in CatSper [81].

## 3. Chloride Channels

The information on the role of chloride channels (and transporters) in spermatozoa is rather scarce when compared to other ion channels. Nevertheless, in sperm cells, they seem to support events such as the hyperpolarization of the plasma membrane [82], the alkalization of cytosolic environments [83,84] and cAMP/PKA-induced protein phosphorylation [84]. All of these processes are typical features of sperm capacitation. The following Cl^−^ channels have been identified in sperm cells: cystic fibrosis transmembrane conductance regulator (CFTR) [82], the γ-aminobutyric (GABA)-gated and related glycine-gated neurotransmitter receptors [85], Ca^2+^-activated Cl^−^ channels (CaCCs) [86], and ClC-3 channels [87]. In this paper, CaCC, ClC-3, and CFTR are discussed.

### 3.1. Ca^2+^- Activated Cl- Channels (CaCCs)

Calcium-activated chloride currents were first noted, in 1981, in *Rana pipens* eggs [88,89]. Ca^2+^-activated Cl^-^ channel (CaCC) opening is stimulated by increases in [Ca^2+^]_i_ concentrations according to release from intracellular stores or influx through plasma membrane channels. CaCCs are anionic channels and belong to the anoctamin family (ANO/TMEM16). The anoctamin family is composed of 10 members, ANO1–ANO10, encoded by the *TMEM16A–TMEM16K* genes, respectively; however, only the products of TMEM16A and TMEM16B express functional CaCCs [90,91,92,93,94]. The sequence identities between the TMEM16 family members in the transmembrane regions reach maximum values of 50–60%. Topologically, TMEM16 proteins consist of 8 to 10 TMs and cytosolic N- and C-termini. The sequence identity between human and mouse orthologs of TMEM16A is high (91%) and the predicted rat homolog contains more than 123 amino acids at the N-termini. What is interesting is that the TMEM16 family does not share significant sequence and structural similarities to other families of ligand-gated ion channels [92]. The difference lies in the direct interaction of TMEM16A with the pore region and changes in the electrostatic properties of the pore [95]. It is suggested that ANO1 specifically forms a homodimer and is organized as relatively stable and noncovalent assemblies of two identical subunits. A scheme of the TMEM16A structure is presented in Figure 4.

In 2012, Orta et al. published evidence for the presence of CaCC in the heads of mature human spermatozoa, using a patch clamp strategy. They described the biophysical characteristics and suggested that TMEM16A may contribute to Ca^2+^-dependent Cl^-^ currents and may participate in acrosome reactions [94]. Cordero-Martinez et al. [86] immuno-localized TMEM16A at the apical part of the acrosome and in middle piece of the flagella of non-capacitated and capacitated guinea pig spermatozoa. The results indicated that TMEM16A plays an important role in capacitation and takes part in acrosomal reactions, as well exhibiting a potential role in sperm motility, especially in the acquisition of hyperactivated motility. Flagellar hyperactivation is crucial for spermatozoa, to drive through the viscous environment of the oviduct [99]. The inhibition of TMEM16A using blockers affects sperm physiology by inhibiting progressive motility and the acquisition of hyperactivated motility. At the same time, the abnormal increase in [Cl^−^]_i_ and the reduction of [Ca^2+^]_i_ levels were observed. According to the authors, proper concentrations of these both ions are required for adequate capacitation [86]. The participation of CaCC in sperm motility was observed earlier in sea urchins, by Wood et al. [100]. The researchers used niflumic acid, an inhibitor of CaCC, to alter the kinetics of the Ca^2+^ fluctuations and observed gross changes in sperm motility patterns [100]. Despite all the data supporting the role of the CaCC in sperm motility, the exact mechanisms regulating the channel have not yet been determined.

### 3.2. Chloride Channel-3 (ClC-3)

Chloride channels (ClCs) comprise an evolutionary conserved voltage-gated channel family. They are found in bacteria, yeasts, plants, and animals [101,102,103]. The ClC proteins were identified by expression cloning from the electric organ of the marine ray *Torpedo marmorata* with ClC-0 [104]. At present, nine mammalian family members of ClCs are known [105]. Schmidt-Rose and Jentsch [106] experimentally showed that ClC-1 and probably all ClC proteins have 10 or 12 TMs with N- and C-termini residing in the cytoplasm. In mammalian homologs, the N-terminal domain is composed of approximately 50–130 amino acid residues and the amino acid length at the C-terminal domain is approximately 170–420 [107]. ClC-3 is a member of the ClC family, which was first cloned from a rat kidney by Kawasaki et al. [108]. ClC-3, together with ClC-4 and ClC-5, forms a distinct branch of the *ClC* gene family [101]. According to Uniprot KB, murine ClC-3 (encoded by *Clcn3* gene) has 10 TMs and two cystathionine beta-synthase (CBS) domains located on its C-terminus. The expression of ClC-3 produces outwardly rectifying Cl- currents that are inhibited by protein kinase C (PKC) activation [109]. According to some authors [108], ClC-3 is most probably an intracellular voltage-dependent electrogenic 2Cl^-^/H^+^-exchanger.

ClC-3 was detected to be localized in the flagellum of human and rhesus monkey spermatozoa [110,111], and in bovine sperm cells, the channel was additionally detected in the acrosome and midpiece [112]. Nevertheless, according to Liu et al. [87], an activation of chloride channels in the sperm neck may be a key factor in the regulation of sperm motility. According to the authors [87], chloride channels play important roles in the regulation of sperm volume and motility. ClCs-3 have been found in bovine epididymal spermatozoa [113]. Western blot analysis has revealed the presence of ClC-3 at 87 kDa from sperm proteins in tested samples. This channel was expressed in normal sperm cells and asthenozoospermic samples derived from patients [110]. Spermatozoa from men with asthenozoospermia (reduced sperm motility) revealed lower cell volume regulating capacity and mobility and were connected with lower expression levels of ClC-3, in comparison to normal spermatozoa [87]. Interestingly, a study of Smith et al. [111] demonstrated that ClC-3 can bind PP1γ2, which is a key enzyme involved in regulation of sperm maturation and motility.

### 3.3. Cystic Fibrosis Transmembrane Conductance Regulator (CFTR)

The cystic fibrosis transmembrane conductance regulator (CFTR) is a unique ATP-binding cassette (ABC) transporter that functions as an ion channel [114]. It mediates the transport of Cl^-^ and HCO_3_^-^ in an electrochemical gradient in contrast to other ABC transporters that transport substrates against their chemical gradients [115]. Regarding topology, CFTR has 12 TMs organized in two membrane spanning domains (MSD1 and MSD2) connected by a long cytoplasmic loop. The N-terminal lasso motif and C-tail are oriented cytoplasmatically. There are also two nucleotide binding domains (one in the connecting cytoplasmic loop, NBD1, and one in the C-tail, NBD2) and a regulatory domain (amino acids 637–845) containing multiple phosphorylation sites (for PKA or PKC). The binding of ATP to the NBDs and the phosphorylation of the regulatory domain opens the channel [116,117] via a pore formed between TMs 4 and 6 in the cytosol and between TMs 1 and 6 at the extracellular surface [117].

CFTRs have been reported to be present in mammalian spermatozoa [118,119,120,121]. The role of CFTRs in sperm motility is not insignificant. A specific inhibition of this channel has been shown to reduce an increase in pH_i_, cAMP production, and membrane hyperpolarization [121], while, in human spermatozoa, CFTRs are thought to remove Cl^-^ from cells upon capacitation [122]. In mice and guinea pigs, CFTRs are reported to transport Cl^-^ to spermatozoa [120,123]. In addition, CFTRs have also been documented to cooperate with a HCO_3_^-^ transporter of the SLC26 family, SLC26A3, in mouse spermatozoa [123]. Spermatozoa from mice lacking *Slc26a3* have been shown to be completely immotile; however, this deficiency was probably due to severe morphological abnormalities [124]. Similar observations have been made with humans, where males with a defect in SLC26A3 have been reported to suffer from oligoasthenoteratozoospermia (reduced sperm count, motility, and morphology) [125].

## 4. Potassium Channels

Generally, K^+^ channels in spermatozoa are responsible for plasma hyperpolarization, which is required for the initiation/hyperactivation of sperm motility. Several types of motility-related K^+^ channels in sperm cells have been reported, e.g., inwardly rectifying K^+^ (Kir) channels [126,127], voltage gated potassium channels (K_v_1.1) [128], SLO K^+^ channels [129,130,131], and cyclic nucleotide-gated channels (CNGK channels) [132,133,134]. Two diverse K^+^ channels, the SLO K^+^ and the CNGK channels, are discussed below, as they are reported to function in two different groups of organisms, namely mammals and aquatic animals.

### 4.1. SLO K^+^ Channels

So far, SLO1 and SLO3 are the most frequently described K^+^ channels in spermatozoa and are likely primary channels for regulating K^+^ currents. They belong to voltage-gated K^+^ channels encoded by the *SLO* gene family, and their role is the regulation of osmolality and Vm within sperm cells [130]. They are activated by membrane depolarization, [Ca^2+^]_i_ and [Mg^2+^]_i_ [135,136]. The full names of the SLO1 and SLO3 channels are calcium-activated potassium channel (in humans encoded by *KCNMA*, alias *SLO*) and potassium channel subfamily U member 1 (encoded by *KCNU1*, aliases *KCNMA3* and *SLO3*), respectively. SLO3 is referred to as the main K^+^ channel in mouse [130] and human spermatozoa [137]. The channels are responsible for K^+^ efflux, which initiates cell membrane hyperpolarization. The control of Vm, SLO1, and SLO3 is thus involved in the regulation of voltage-sensitive ion channels like VGCC and CatSper. Indeed, an increase in K^+^ in an extracellular environment induces an increase in [Ca^2+^]_i_ in human spermatozoa, along with the addition of progesterone [138]. SLO1 (also known as the big calcium (BK) channel) is expressed across all metazoa and mainly in neural and muscular cells, whereas SLO3 is unique to mammalian spermatozoa [135]; however, exploratory genomics has revealed the presence of the *Slo3* gene in fish, birds, and reptiles [139], and recent findings indicate the presence of a cytoplasmic truncated SLO3 isoform in mouse tissues, such as brain, kidneys, and eyes [140]. Sequence analysis suggests that SLO3 evolved from SLO1 via gene duplication, and that, in humans, the sequences feature a 42% identity [135]. The differences in the sequences contribute to functional diversity of these two channels. Although both of them are voltage-gated, SLO1 is additionally activated by Ca^2+^ [141], whereas SLO3 is activated by an alkaline pH_i_ [135,142]. Both channels are formed by four pore-forming α subunits and several auxiliary subunits (Figure 5) [135,143].

The fast evolution of SLO3 has also resulted in a high degree of structural divergence and various functional properties for this channel within mammals. The comparison of SLO3 channels from bovine and mouse spermatozoa indicates prominent length and sequence polymorphisms within the RCK1 (regulator of K^+^ conductance 1) domain, resulting in different voltage ranges for activation and kinetics [145]. Accordingly, the differences in the amino acid sequences can be a reason underlying the different sensitivity of human SLO3 compared to murine SLO3. According to Brenker et al. [129], human SLO3 is activated weakly by pH_i_ and more strongly by Ca^2+^, and therefore Vm is more strongly regulated by [Ca^2+^]_i_ than by pH_i_. Going further, the authors have proposed that progesterone-induced Ca^2+^ influx via CatSper is limited by Ca^2+^-controlled hyperpolarization via SLO3. A recent study of Geng et al. [137] indicated that SLO3 is evolving significantly more rapidly in humans than SLO1. Additionally, there is a single nucleotide polymorphism that changes an amino acid side chain in RCK1 at a site of interface between the cytoplasmic gating ring of the channel and the intramembrane-gated pore. This variant allele (C382R) confers heightened sensitivity to both Ca^2+^ and pH_i_ with the SLO3 channel [137]. It is noteworthy that the murine SLO3 channel has been documented to have a high affinity towards PIP2, which is a tiny phospholipid that regulates a variety of physiological processes [146]. The binding of PIP2 to SLO3 upregulates the channel’s activity, whereas the dephosphorylation of PIP2 by a voltage-sensing phosphatase (VSP) makes SLO3 less active. It seems that all of these events of SLO3 regulation are independent of the presence of the Lrrc52 (Leucine-rich repeat-containing membrane protein 52) auxiliary subunit [131,147].

Like most of the motility-related sperm channels, SLO3 is localized within a principal piece of the sperm flagellum [129]. Mice lacking SLO3 have been shown to produce spermatozoa with reduced motility that are incapable of fertilization under in vitro conditions [148]. The reduced sperm motility in *Slo3*-/- mice may be a result of a lack of membrane hyperpolarization [130] and therefore a lack of activation of CatSper and other voltage-gated channels. The spermatozoa of SLO3-deficient mice have also been reported to display abnormal morphology, probably due to a diminished regulation of osmotic homeostasis [130]. Accordingly, as Lrrc52 and SLO3 are co-expressed [143], the genetic knockout of Lrrc52 in mice severely reduces fertility, and sperm cells lacking Lrrc52 require greater positive voltages and pH levels to induce a K^+^ current, compared to spermatozoa from wild-type mice [149]. In humans, ca. 10% of sub-fertile patients undertaking in vitro fertilization (IVF) and intracytoplasmic sperm injection (ICSI) have a negligible K^+^ outward conductance or an enhanced inward conductance in their spermatozoa, thereby resulting in the depolarization of sperm Vm [150].

### 4.2. Cyclic Nucleotide-Gated K^+^ Channel (CNGK)

In aquatic animals, other types of K^+^ channels are reported to be responsible for the regulation of [K^+^]_i_. Regarding ionic milieu, K^+^ concentrations in fresh water are very low, whereas, in seawater and oviducts, they are extremely high [151,152]. Therefore, the mechanisms regulating [K^+^]_i_ are different. In aquatic animals, the hyperpolarization of the sperm plasma membrane is evoked by the activation of cyclic nucleotide-gated K^+^ channels (CNGKs) which are responsible for the K^+^ efflux. CNGK channels are found in sea urchin [134,153] and zebrafish [132] spermatozoa and generally their activation is triggered by the presence of oocyte-derived chemoattractants. Topologically, a CNGK channel is a tetrameric channel created by four homologous repeat sequences. These four sequences, from 1 to 4, are connected in the cytoplasm with their N- and C-termini and only the N-terminus of sequence 1 and the C-terminus of sequence 4 are located freely in the cytoplasm. CNGK also has four cyclic nucleotide binding domains (CNBDs), with three in the cytoplasmic loops between the sequences 1–2, 2–3, and 3–4 and one in the C-tail of sequence 4. Each sequence consists of six TMs; TM4 is voltage-sensitive, and a loop between TM5 and TM6 creates a pore [154]. The signal from a chemoattractant (i.e., speract) is translated into an increase in cGMP (cyclic guanosine monophosphate, e.g., in sea urchin *Arbacia punctulata*, one speract moiety generates ca. 45 cGMP molecules [153]). Moreover, cGMP opens CNGK channels, to produce transient membrane hyperpolarization, followed by a sustained depolarization in sea urchin spermatozoa [134,153]. Additionally, sea urchin CNGKs were documented to also be activated by Zn^2+^ [133]. Nevertheless, Fechner et al. [132] documented that zebrafish CNGK orthologs are activated by alkaline pH_i_ and not cGMP. The localization of CNGK is different in spermatozoa from these two species, where, in sea urchins, the channel was detected in the flagellum [134], whereas, in *Danio rerio*, it was detected in the sperm head [132]. These findings show that sperm signaling among aquatic species displays unique variations, which probably represent adaptations to vastly different ionic environments and fertilization habits. The role of CNGK channels in the induction of sperm motility is essential, as their activation leads to Ca^2+^ entry mediated by voltage-sensitive Ca^2+^ channels [132,133,134].

The regulation of cytosolic K^+^ is also regulated by Na^+^/K^+^-ATPase (NKA), which imports K^+^ into the cell. NKA generates a Na^+^ gradient across the plasma membrane, thereby providing the energy required for normal function of the Na^+^/H^+^-exchanger involved in the alkalization of the pH_i_ in spermatozoa [155].

## 5. Voltage-Gated Na^+^ Channels (VGNCs, NaV)

Voltage-gated Na^+^ channels (VGNCs) also regulate sperm motility. These channels are activated when the hyperpolarized plasma membrane becomes depolarized (−70 mV to −55 mV to 0 mV) and conduct Na^+^ inward the cell. In excitable cells like neurons and myocytes, VGNCs are involved in the propagation of action potential. A single VGNC consists of a large α subunit which forms a core of the channel. The VGNC channel family has nine members, named Na_v_1.1–Na_v_1.9, which are encoded by nine different genes (*SCN1A–SCN9A*, respectively). Additionally, a tenth non-voltage-gated atypical α isoform has been identified, namely Na_x_ (a product of the *SCN7A* gene) [156]. The amino acid identity between members is >50%, and the differences are related to the tissue expression, sensitivity, and kinetics of the channels [157]. Alone, the pore-forming α subunit is sufficient for functional expression; however, the kinetics and voltage dependence of the channel gating are modified by auxiliary β subunits. In mammals, there are five β subunits (β1, β1B, β2, β3, and β4, encoded by the four genes of *SCN1B–SCN4B*, respectively), but most Na_v_ members have one or two β subunits that modulate Na_v_ activity. The subunits β2 and β4 bind to the α subunit via a disulfide bond, whereas the β1 and β3 subunits are associated non-covalently. All β subunits are transmembrane proteins, but an exception is β1B, which is a soluble molecule (reviewed in References [157,158]). A VGNC scheme is shown in Figure 6.

VGNCs have been reported to be present in the sperm cells from humans [156,161] and bulls [162]. In human spermatozoa, mRNAs have been detected for all of the Na_v_ members (Na_v_1.1–Na_v_1.9 and Na_x_) and for all auxiliary β subunits [156]. The immunostaining of human spermatozoa has shown that VGNCs are located in the sperm principal piece (Na_v_1.6, Na_v_1.8, and Na_x_), connecting piece (Na_v_1.2, Na_v_1.4, and Na_v_1.7), head (Na_v_1.5), and midpiece (Na_v_1.9) (see References [156,161]). It is very likely that this specific distribution reflexes signaling and metabolic pathways in spermatozoa. A role of VGNCs in sperm cells is the maintenance of progressive motility rather than hyperactivation, although they are also engaged in capacitation through the tyrosine phosphorylation of proteins localized in the post-equatorial region, midpiece, and flagellum [163,164]. In human spermatozoa, pharmacological stimulation of Na_v_1.8 has increased progressive motility but has not induced hyperactivation or an acrosome reaction [161].

## 6. Voltage-Gated H^+^ Channels (VGHCs, H_v_s)

The main role of a voltage-gated H^+^ channel (VGHC, which is encoded by the *HVCN1* gene in humans) is to extrude H^+^ from a cell, leading to an increase in pH_i_. VGHCs have been found in a diverse spectrum of species, ranging from unicellular marine animals to humans, and also within different cell types [165]. The channels are characterized by highly selective H^+^-conductance, opening with membrane depolarization, high extracellular pH and decreased pH_i_ (high ∆pH), strong temperature dependence for both conduction and gating, and an absence of inactivation [165]. Additionally, VGHCs are activated by the endocannabinoid anandamide and via the removal of extracellular Zn^2+^ which is a proton channel blocker [166]. In chordates (*Ciona intestinalis*, mice, and humans), VGHCs are dimers; however, the monomers are also functional as each monomer has its own conduction pathway. Each VGHC monomer consist of only four TMs which create VSDs in other voltage-gated channels. Interestingly, Berger et al. [167] identified an alternative isoform of VGHC in human spermatozoa, namely H_v_1Sper. The isoform is formed from VGHC via the removal of 68 amino acids from the N-terminus during post-translational proteolytic cleavage, which consequently makes H_v_1Sper more sensitive to ∆pH. Moreover, VGHC and H_v_1Sper can form heterodimers. According to the authors, the cleavage and heterodimerization of H_v_1Sper may be an adaptation to the specific requirements for pH control in spermatozoa [167]. A schematic of the VGHC structure is presented in Figure 7.

Besides human spermatozoa [166,167,170,171], VGHC channels have also been found in sperm cells from other mammals, like macaques [172], boars [173], and bulls [33]. According to the literature, VGHCs are localized in the sperm principal piece and co-localize with CatSper channels that are distributed asymmetrically within bilateral longitudinal lines [32,171,174]. The VGHC localization strictly corresponds to their role in sperm physiology. Activation results in intracellular alkalinization, which is followed by the activation of flagellar CatSper, the influx of Ca^2+^, and thus the induction of hypermotility. Indeed, several studies have indicated a functional relationship between VGHCs and CatSper [32,166,170,175]. It has been suggested that the specific distribution of VGHCs along the sperm flagellum provides a structural basis for the selective activation of CatSper and subsequent flagellar rotation [171]. The VGHC has been proven to be involved in progressive sperm motility, capacitation, and acrosome reactions [166,174], and a blockage of VGHCs results in a reduction of sperm motility parameters and a progesterone-induced acrosome reaction [170,173]. Additionally, the VGHC and CatSper channels have been found to be involved in the progesterone-induced generation of reactive oxygen species (ROS) via membrane NADPH oxidase 5 (NOX5) [175]. The binding of Ca^2+^ to NOX5 activates the enzyme for superoxide (O_2_^-^) production [176]. Although an overproduction of ROS is negatively correlated with sperm motility, it must be pointed out that a certain level of ROS is necessary for proper functioning of spermatozoa (revised in References [177,178]).

## 7. Conclusions

The review describes some of the more important ion channels involved in the regulation of sperm motility in different animal species based on the available literature. A clear consensus has emerged from the reviewed literature, where the known ion channels governing sperm motility are mainly voltage-gated channels. In some cases, specific physiological adaptations, such as internal fertilization, have supported the development of novel isoforms (e.g., H_v_1Sper [167]) in evolution, and even novel channels like CatSper [57]. The aim of these evolutionary improvements is to support spermatozoa with efficient machinery to ensure successful egg fertilization. To fulfill this, the channels have to react adequately to changes in Vm, pH_i_, and concentration changes of a spectrum of biomolecules. Some channels (e.g., SLO1, CaCCs, and VGCCs) are regulated by Ca^2+^, which is an important secondary messenger that regulates sperm motility. In some of the ion channels described in this review (Figure 2, Figure 6 and Figure 7), the Ca^2+^-induced activation of the sAC/cAMP/PKA pathway not only phosphorylates proteins related to sperm motility [13], but also the ion channels that regulate their function. Nevertheless, it has to be noted that this precise ionic mechanism is also supported by ion pumps (e.g., PMCA and NKA) and transporters (e.g., NCXs, Na^+^/H^+^ exchangers, and bicarbonate transporters) that support the functions of the channels [23,24,126,160]. It also must be noted that sperm motility also requires energy in the form of ATP. Mitochondria represent the main source of ATP in spermatozoa and any dysfunction of these organelles results in reduced sperm motility [179]. Mitochondria have two membranes (outer and inner) which contain their own specific ion channels, transporters, and pumps that are involved in mitochondrial ion homeostasis, functioning, and the maintenance of mitochondrial membrane potential [16,179]. All of them are known to be engaged in the regulation of sperm motility [16,179] and deserve separate discussion.

In conclusion, understanding the roles of sperm ion channels, their structures, and sperm specificities is a crucial step towards investigating potential implications in the treatment of infertility [150,180] and/or contraception [181].

## Figures and Tables

**Figure 1 ijms-22-03259-f001:**
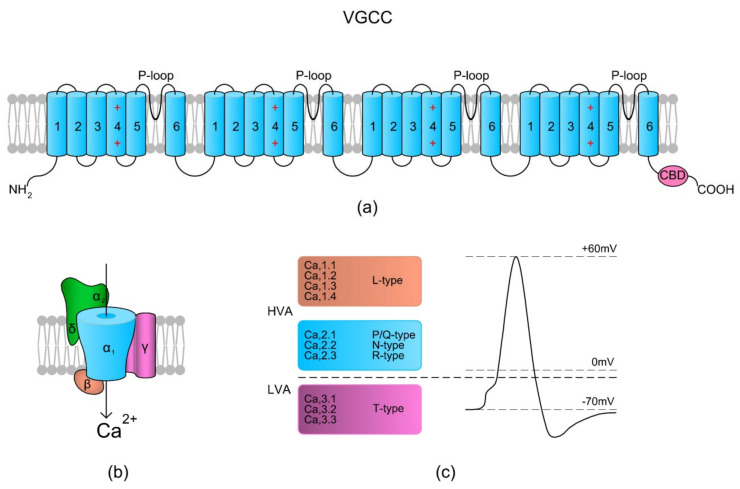
Voltage-gated Ca^2+^ channel (VGCC) structure scheme. (**a**) The topology of the α1 subunit is made up of four homologous domains that each consist of six transmembrane α helices (TM1–6). TM4 from each homologous domain serves as the voltage sensor moving outward and rotates under the influence of the electric field, thereby initiating a conformational change that opens the respective pore. TM5, TM6, and the loop between them (P-loop) from each domain form a pore. The C-terminal tail contains a Ca^2+^ binding domain (CBD) and in some types of VGCCs a site for calmodulin (calcium-modulated protein; CaM) binding. The binding of Ca^2+^ to CBD or via CaM inactivates the channels. (**b**) A schematic presentation of the VGCC subunits (α1, α2δ, β, and γ) with their spatial localizations. (**c**) Overview of the types of VGCCs in relation to Vm-dependent activation – high voltage activation (HVA) and low voltage activation (LVA) (based on References [26,27]).

**Figure 2 ijms-22-03259-f002:**
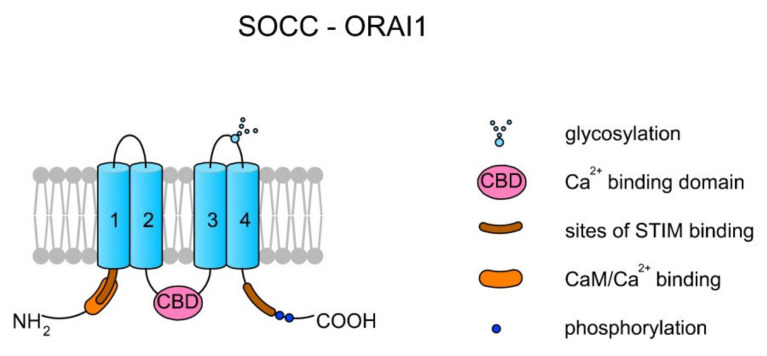
Topology of a store-operated Ca^2+^ channel (SOCC) created by ORAI1. Each ORAI protein has four TMs. TM2 and TM3 create a pore. There are two sites for STIM1 binding at the N- and C-termini. The interaction between STIM1 and ORAI activates the channel and the release of Ca^2+^ from the endoplasmic reticulum (ER). The binding of Ca^2+^ by the Ca^2+^ binding domain (CBD) localized on the central loop inactivates the channel [51]. Additionally, it can also be inactivated upon CaM binding [52].

**Figure 3 ijms-22-03259-f003:**
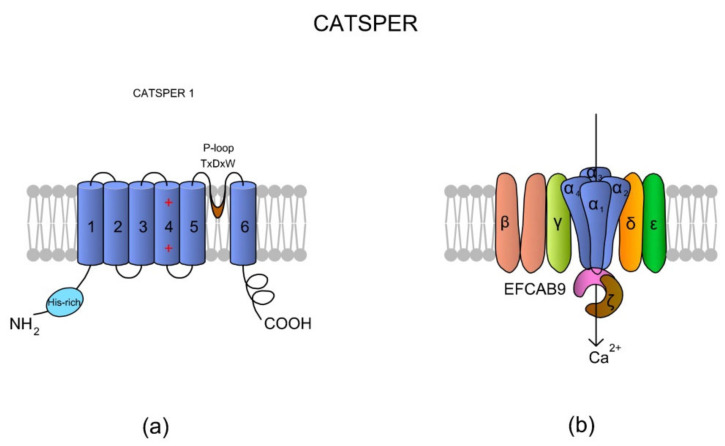
A topological and spatial structure of CatSper. (**a**) The α1 subunit created by CatSper1. Like most voltage-gated channels, each α subunit contains six transmembrane domains (TM1–TM6) creating two physiologically distinctive regions, namely the voltage-sensing domain (VSD; TM1–4) and pore-forming region (TM5–6). Each TM4 contains several (two to six) positively charged amino acid residues that serve as voltage sensors (reviewed in Reference [57]). Voltage slopes move TM4, resulting in conformational changes that open and close the channel pore [64]. Additionally, a short and hydrophobic cyclic structure linking TM5–6 contains a conserved homologous amino acid sequence (T × D × W), which selectively permits Ca^2+^ influx. The N-terminus of CatSper 1 contains a specific histidine-rich region that might be involved in the pH regulation of CatSper activity. (**b**) The topological localizations of all auxiliary subunits are not randomly organized. The auxiliary CatSperβ subunit has two predicted TMs that are separated by a large (ca. 1000 amino acids) extracellular loop [64], whereas CatSperγ, CatSperδ, and CatSperε feature only one TM. Brown et al. [69] suggested that CatSperζ is a late evolutionary adaptation to maximize fertilization success inside the female mammalian reproductive tract. The predicted topology of Hwang et al. [62] situates the CatSperζ and EFCAB9 subunits as a cytoplasm complex that is located just below the CatSper 1–4 subunits. This complex interacts with the channel pore as a gatekeeper. The increase in pH_i_ causes Ca^2+^ binding to highly conserved EF-hands of EFCAB9, leading to dissociation of the EFCAB9-CatSperζ complex and full activation of the channel. Accordingly, EFCAB9-CatSperζ appears to be responsible for both modulation of the channel activity and organization of the CatSper domains [62]. The scheme has been prepared based on Reference [62].

**Figure 4 ijms-22-03259-f004:**
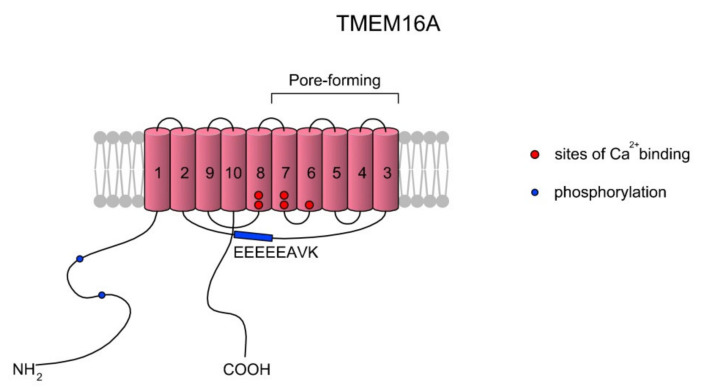
A simplified topology of the TMEM16A monomer. Each monomer has 10 TMs. The ion conduction pore of TMEM16A is formed by TMs three to seven in each subunit, and thus the CaCC features two pores [96,97]. As summarized in a review of Ji et al. [97], the activation of TMEM16A is gated by two main mechanisms: voltage (Vm) and low concentrations of Ca^2+^ (<600 nM) via the EEEEEAVK motif in the TM2–TM3 loop. Contreras-Vite et al. [98] proposed a gating mechanism model where TMEM16A is directly activated by the Vm-dependent binding of two Ca^2+^ ions coupled by a Vm-dependent binding of one external Cl^−^ ion. The scheme was prepared based on Reference [97].

**Figure 5 ijms-22-03259-f005:**
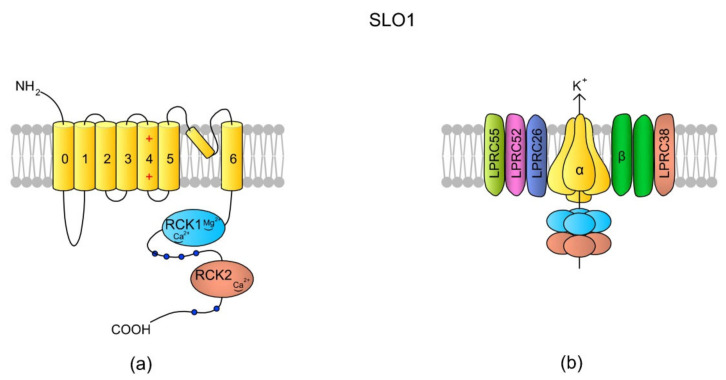
SLO1 structure scheme. (**a**) A topology of the α subunit. Each α subunit consists of seven (0–6) TMs, where TM4 is a typical voltage-sensing domain (VSD). An extracellular loop between TM5 and TM6 forms the pore. The N-tail is located extracellularly but the C-end is a long tail containing the RCK1 (regulator of K^+^ conductance 1) and RKC2 domains [135]. The structural difference between SLO1 and SLO3 is that there are “Ca^2+^-bowl” structures within the RKC domains of SLO1, making the channel sensitive to [Ca^2+^]_i_. (**b**) In the tetrameric structure of the channel, the cytoplasmic C-termini creates a gating ring. According to the literature, SLO1 has five auxiliary subunits: one β subunit (with two transmembrane domains) and four Leucine-rich repeat-containing membrane proteins (LRRCs, also named γ subunits), LRRC26, LRCC52, LRRC55, and LRRC38, which modulate SLO1 sensitivity to Vm and [Ca^2+^]_i_ (revised by Reference [144]). In murine testes and spermatozoa, two auxiliary subunits of the SLO3 channel have been identified: Lrrc52 and Lrrc26. Both of them are involved in the regulation of SLO3, and the expression of Lrrc52 is critically dependent on the presence of SLO3 [143]. The schemes are adapted from References [136,144].

**Figure 6 ijms-22-03259-f006:**
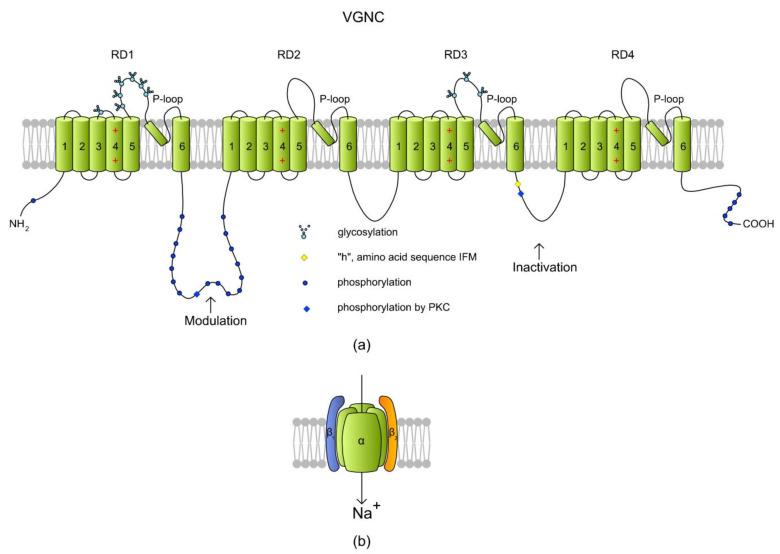
A structure of a voltage-gated Na^+^ channel (VGNC) based on a SCN2A isoform. (**a**) The α subunit is created by four repeat domains (RD1–RD4) that each have six TMs. Classically, TM1–TM4 of each domain form a VSD where TM4 acts as a positively charged sensor. During depolarization, TM4 is believed to move toward the extracellular surface, allowing the channel to become permeable to ions. Na^+^ is transported inside a cell through a pore (P-loop) formed between TM5 and TM6 of each RD. The RDs are connected with long intracytoplasmic loops with sites for protein phosphorylation via PKA and PKC [157]. The cytoplasmic loop between RD3 and RD4 contains an “h” (I × F × M sequence) motif, which stands for a hydrophobic triad of amino acids, namely, isoleucine, phenylalanine, and methionine (I1488, F1489, and M1490). The IFM motif is involved in the inactivation of VGNC, serving as a hydrophobic latch for a hinged lid formed by the loop between RD3 and RD4 [159]. Phosphorylation in the RD1/RD2 and RD3/RD4 loops modulates the channel inactivation (adapted from Reference [157], revised in Reference [160]). (**b**) A cartoon of VGNC created by the pore-forming α subunit and the two auxiliary β subunits.

**Figure 7 ijms-22-03259-f007:**
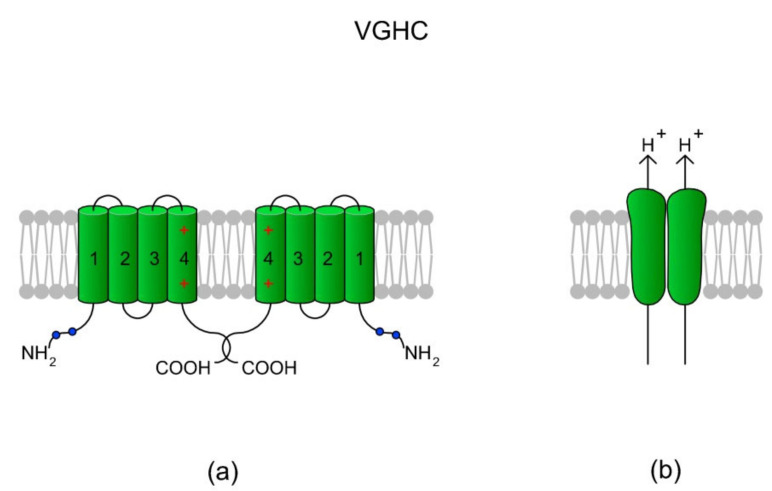
A voltage-gated H^+^ channel (VGHC) and its structure. (**a**) A VGHC monomer is created by four TMs which in classical voltage-gated channels comprise VSD. Accordingly, VGHCs do not possess a pore-forming domain (TM5-TM6) and the extrusion of H^+^ ions probably takes place via a water wire spanning the VSD [168]. According to Boonamnaj et al. [169], in VGHC dimers, C-terminal tails interact by forming a coiled structure that stabilizes the channel. Sites of phosphorylation in the N-termini may enhance the selectivity of the channel. (**b**) A dimeric structure of a VGHC. As the VGHC has no pore-forming domains, H^+^ diffuses through each monomer.

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
