# Peer review of "Structure and Function of Ion Channels Regulating Sperm Motility—An Overview"

_ijms, 2021, doi:10.3390/ijms22063259_

Round 1
Reviewer 1 Report
The review on the "Structure and function of ion channels regulating sperm motility" (please note how I have worded the title) is a timely and admirable overview that is currently needed in this field. The logical arrangement of the subject is acceptable. However, the readability is to a large extent obscured by the obtuse handling of sentences and terminologies. Therefore my first comment is for the authors to get the manuscript read and fixed by an English-speaking/writing person. Other than that I am extremely delighted by this review article and once the readability is improved I would wholeheartedly approve of its publication. Additionally, I have the below comments, mostly related to the readability and a couple of other matters.
The review article would benefit from a page of abbreviations. There are quite a few abbreviations such as cAMP, CaM, ER, TM, that seem to suddenly pop up in the article without an expanded version ahead of it or with an expanded version only in the legends of the figures.
lines 20, 21 - These sentences would read better in the active voice. Ex: The role of other ion channels in the spermatozoa, such as x, y, and z, is to ensure the activation and modulation of CatSper.
Multiple locations: May I advise the use of the term motility rather than movement with reference to the spermatozoa motility.
Also, as much as possible please do use the term spermatozoa rather than the colloquial 'sperm'.
line 35: "Generally, in marine fishes, the spermatozoa motility is induced..."
line 42: instead of 'just' use, 'immediately' or 'soon'
Line 43: In mammals, where the fertilization is internal, the spermatozoa is activated during their transit...
Line 57: Proper balance of intracellular ions
Line 83 could be reworded such as: In this review, the structure and physiological role of some of the important ion channels that are critical to spermatozoa motility are discussed.
Line 86: As mentioned above, the Ca2+ channels play an important role in the regulation of sperm motility as Ca2+ is a common secondary messenger engaged in several cellular signaling pathways as well. Ca2+ is required for sAC/cAMP/PKA pathway activation [10], as well as the maintenance of sperm mitochondrial functioning and ATP production [16] which are critical for sperm cell motility.
Line 154: ... are more frequently reported.
Sentence 154-7: modify the sentence.
Sentence 159: "size of the current" is improper quantification of electricity. Please rewrite this sentence.
Lines 156, 172, etc. Scientific nomenclature (Labeo rohita, Salmo salar, Arbacia punctulata, Danio rerio, etc.) must be italicized. Please check the journal requirements.
Line 168: In a study by Toni et al....
Line 175: not only activates basic and hyperactivate motility
Line 176: Immunostaining studies on TRPV1 indicates that ...
Above referred to sentence needs modification as it implies that TRPV1 is present everywhere on spermatozoa.
Sentence beginning in 178: "While TRPV1is activated by x,y, and z, TRPV4 is ...". Authors may also simply delete the word "next" in line 178.
Line 182: Instead of the term "indicated" use "demonstrated" or "showed"
Line 186: Trpv4 to caps.
Line 216: by displacing
Sentence in 456 needs to rewritten by the addition of descriptive terms between 'depolarization' and [Ca2+]
Line 497: Mice lacking SLO3 have been shown to produce spermatozoa with reduced motility that are incapable of fertilization.
Line 524: In aquatic animals, ...
Line 217: It could be suggested that...
Line 217: "According to..." what? Did the authors mean to say based on the understanding from the previous statement?
Line 231: Orai1 -> ORAI1
Sentence 256 needs modification.
Line 606: and via the removal of extracellular Zn2+ which is a proton channel blocker [168].
Line 615: A schematic of the VGHC structure is shown in Figure 7.
Line 626: delete "data"
Line 631: Indeed, several studies have indicated...
Line 629: ...results in increased intracellular alkalinization which is followed by...
Sentence 645: The review describes some of the more important ion channels involved in ... in different animal species based on the available literature.
Line 648: In some cases, specific physiological adaptations such as internal fertilization...
Line 655: In some of the ion channels described in this review (Fig. 2, 6, 7)...
Line 662: ... must be noted...
Author Response
Dear Reviewer,
Thank you very much for your valuable comments regarding our manuscript.
We have introduced all the recommended corrections including:
- Corrected spelling of the title;
- sperm -> spermatozoa and movement -> motility transitions, where possible;
- A list of abbreviations at the end of the manuscript was added;
- Italicized latin names of species and names of genes;
- Corrected or rewritten sentences.
All the corrections suggested by the Reviewer have been marked with red colour and are visible in the text.
It also has to be added that the manuscript was subjected to English editing at MDPI service and a certificate of such edition was obtained.
Reviewer 2 Report
Dear authors, I really appreciate your effort.
The manuscript collects and resumes most relevant knowledge about the different types of ion channels involved in the control of various type of sperm motility. It also gives detailed information on the various molecular structures and molecular differences of species. The bibliography is extremely numerous, and the citations in the text are all justified and discussed. In my opinion it is an excellent work, interesting for both molecular biologists and reproductive physiologists.
The manuscript is very interesting and informative, well written and extensively collects and compares contributions on the subject with an excellent evaluation among species. In my opinion it can be published just as it is.
Author Response
Dear Reviewer,
Thank you very much for such comments. We do strongly believe the manuscript will be useful for biology-related scientists.
Reviewer 3 Report
This is a correct bibliographic review. That addresses the theme expressed in its title.
The bibliography used is considered current. And given the complexity of the subject, an appropriate text structure has been used. Perhaps some specific data on domestic animals such as sheep, goats, bulls and pigs could be added.
Author Response
Dear Reviewer,
Thank you very much for your comments and suggestions.
We would like to add more information regarding ion channels regarding domestic animals. Indeed, we have added just one full sentence and some short additional information to the text (marked with purple color) however, it was difficult to find other similar examples. The found papers either indicate changes in ion current upon stimulation with different ionophores/cryopreservants not indicating which exact channel is stimulated (mainly papers printed 10 and more years ago), and/or are not relevant to this manuscript. This shows how little studies is carried out on domestic species.
It was our intension to compare data from different animal species. In our paper we have already included information on sperm ion channels from domestic animals such as bulls, pigs and chickens (examples are marked with purple color). Taking into account fertility problems, we hope our manuscript will serve as a source of knowledge for physiologists working on reproductive biology of all animals, domestic and wild ones.